# Heterologous vaccination regimens with self-amplifying RNA and adenoviral COVID vaccines induce robust immune responses in mice

Alexandra J. Spencer [1]✉, Paul F. McKay [2], Sandra Belij-Rammerstorfer[1], Marta Ulaszewska[1], Cameron D. Bissett[1], Kai Hu [2], Karnyart Samnuan[2], Anna K. Blakney [2], Daniel Wright [1], Hannah R. Sharpe [1], Ciaran Gilbride[1], Adam Truby[1], Elizabeth R. Allen [1], Sarah C. Gilbert [1], Robin J. Shattock [2] & Teresa Lambe [1]

Several vaccines have demonstrated efficacy against SARS-CoV-2 mediated disease, yet there is limited data on the immune response induced by heterologous vaccination regimens using alternate vaccine modalities. Here, we present a detailed description of the immune response, in mice, following vaccination with a self-amplifying RNA (saRNA) vaccine and an adenoviral vectored vaccine (ChAdOx1 nCoV-19/AZD1222) against SARS-CoV-2. We demonstrate that antibody responses are higher in two-dose heterologous vaccination regimens than single-dose regimens. Neutralising titres after heterologous prime-boost were at least comparable or higher than the titres measured after homologous prime boost vaccination with viral vectors. Importantly, the cellular immune response after a heterologous regimen is dominated by cytotoxic T cells and Th1$^+$ CD4 T cells, which is superior to the response induced in homologous vaccination regimens in mice. These results underpin the need for clinical trials to investigate the immunogenicity of heterologous regimens with alternate vaccine technologies.

[1] Nuffield Department of Medicine, The Jenner Institute, University of Oxford, Oxford, UK. [2] Department of Infectious Disease, Imperial College London, London, UK. ✉email: alex.spencer@ndm.ox.ac.uk

A number of vaccines against severe acute respiratory syndrome coronavirus 2 (SARS-CoV-2) have reached late-stage clinical trials with encouraging efficacy readouts and mass vaccination schemes have been already initiated across several countries. Multiple vaccine technologies are being advanced[1], but there is limited information on how these vaccine modalities may work in combination. With ongoing clinical trial assessment of numerous SARS-CoV-2 vaccines and mass vaccination incentives, there is a recognized real-world scenario where individuals may be vaccinated with different vaccine modalities. However, the utility of vaccination regimens combining different types of vaccine approaches remains to be determined.

Self-amplifying RNA (saRNA) typically encodes the alphaviral replicase and a target antigen. Upon entry into the cytoplasm, the RNA is amplified with subsequent translation of the target antigen[2]. The SARS-CoV-2 saRNA vaccine has been demonstrated to be highly immunogenic in preclinical animal models[3] and is progressing through clinical trial assessment with promising results. Adenoviruses are a frequently used viral vector vaccine technology, they can be rapidly made to clinical grade at large scale and a single vaccination can be sufficient to provide rapid immunity in individuals[4]. In particular, chimpanzee-derived adenoviruses (ChAd) have good safety profiles, while inducing strong cellular and humoral immune response against multiple target disease antigens[5–7]. Previous studies have demonstrated efficacy after vaccination with ChAdOx1 nCoV-19/AZD1222 against SARS-CoV-2-mediated disease[8], with concomitant high-titre humoral immune responses and a measurable Th1-dominated cellular immune response[9,10]. Importantly, an immune profile was consistently observed across different animal species[11,12].

In this study, the immunogenicity of saRNA and ChAdOx1 vaccines expressing full-length SARS-CoV-2 spike protein were assessed in mice following vaccination with different combinations of vaccine modalities. We demonstrate robust antibody responses following heterologous vaccination regimens, with high-titre neutralizing antibodies. The cellular immune response is dominated by cytotoxic T cells secreting interferon-γ (IFNγ) and tumor necrosis factor-α (TNFα), and antigen-specific CD4+ T cells of a Th1 phenotype, with significantly higher antigen-specific responses observed following heterologous vaccination than those responses induced in single-vaccine regimens. These results underpin the need for clinical trial assessment of immunization regimens with alternate vaccine modalities.

## Results

**Heterologous vaccination potentiates SARS-CoV-2 spike-specific antibody response.** It was previously shown that homologous prime-boost immunization with saRNA[3] or ChAd[11] induced a SARS-CoV-2 spike-specific IgG response with neutralization capacity. In this series of experiments, we compared the immune response induced by heterologous, homologous and single-dose vaccination with ChAd and saRNA vaccine modalities.

Heterologous vaccination with either ChAd or saRNA prime and alternative boost (i.e., ChAd-saRNA or saRNA-ChAd) or two doses of saRNA induced the highest IgG responses, compared to mice vaccinated with a single dose of either ChAd or saRNA (Fig. 1A and Supplementary Fig. S1A), with a strong correlation between the two independent IgG enzyme-linked immunosorbent assay (ELISA) methods (Supplementary Fig. S1B). This IgG response showed a mixed profile, with mainly IgG2 (IgG2a, IgG2b in both mouse strains and IgG2c in CD1 mice) and IgG1 subclasses (in both mouse strains) (Fig. 2A, B), which was similar to previously reported results after vaccination with ChAd alone[13] or saRNA alone[3].

Comparable amounts of SARS-CoV-2 spike-specific IgM were detected in all vaccinated groups (Fig. 1B), whereas heterologous vaccination with saRNA-ChAd induced higher serum SARS-CoV-2 spike-specific IgA levels compared to single vaccination with either vaccine, in both CD1 and BALB/c mice (Fig. 1B).

Two doses of vaccine were also shown to increase the avidity of the IgG towards SARS-CoV-2 spike (Fig. 1C) compared to single administration of either vaccine, whereas antibody-mediated neutralization was measured across all groups of vaccinated mice, with heterologous and two-dose saRNA inducing significantly higher levels of neutralization compared to single-dose vaccination regimens (Fig. 1D). Not surprisingly, a correlation was measured between neutralization and levels of IgG /IgA SARS-CoV-2 spike-specific antibodies (Supplementary Fig. S1C).

Flow cytometry staining enabled identification of SARS-CoV-2 spike-specific B cells in the spleen of BALB/c mice. The data demonstrated that all vaccination regimens induced a similar number of antigen-specific B cells (Supplementary Fig. S2B), with similar numbers of germinal centre (GC) and isotype class-switched B cells observed in all groups of vaccinated mice. Formation of GC is important for generation of long-lived memory B cells and the data demonstrate that changing vaccine modalities did not have an impact on the formation of GC B cells.

**Heterologous vaccination induces strong Th1-type response.** T-cell-mediated immunity was also investigated following heterologous or homologous ChAdOx1 and saRNA vaccination regimens. The highest IFNγ response detected by enzyme-linked immune absorbent spot (ELISpot) was observed in mice that received a heterologous combination of vaccines; this increase was only statistically significant when compared to a single administration of saRNA in both strains of mice (Fig. 3A, B) or saRNA-ChAd compared to ChAd in BALB/c mice (Fig. 3B). In agreement with earlier reports, ELISpot responses were primarily directed towards the S1 portion of the spike protein (Supplementary Fig S3A), with a consistent breadth of response measured with all vaccine combinations and in both strains of mice (Fig. 3A, B).

Phenotype and functional capacity of the T-cell response was measured by intracellular cytokine staining, together with memory T-cell marker staining. Consistent with previously published data, saRNA and ChAd vectors induced antigen-specific CD4+ T cells of a Th1 bias, with minimal interleukin-4 (IL4) and IL10 production measured, and a response dominated by the production of IFNγ and TNFα, regardless of the vaccination regimen (Fig. 4A). Antigen-specific CD4+ T cells displayed mixed T effector (Teff) and Teff memory (Tem) phenotype, with no statistically significant differences observed in the total number of cells (Fig. 4A) and cell subsets (Supplementary Fig. S3C). As before, in mice, the cell-mediated response following vaccination was dominated by CD8+ T cells, with much higher levels of IFNγ and TNFα (in addition to upregulation of CD107a) seen in all vaccine groups (Fig. 4B) compared to CD4+ T-cell responses. In addition, the highest number of antigen-specific T cells, in both outbred and inbred strains of mice, was measured following heterologous vaccination, regardless of vaccine order (Fig. 4B), with antigen-specific CD8+ T cells displaying a predominantly Teff phenotype.

## Discussion

The current pandemic and extraordinary efforts to develop effective vaccines with subsequent mass vaccination roll-out has highlighted the 'real-world' practicalities of global vaccination campaigns. There are ~20 vaccines in Phase 3 clinical trial assessment and several vaccines have already reported efficacy

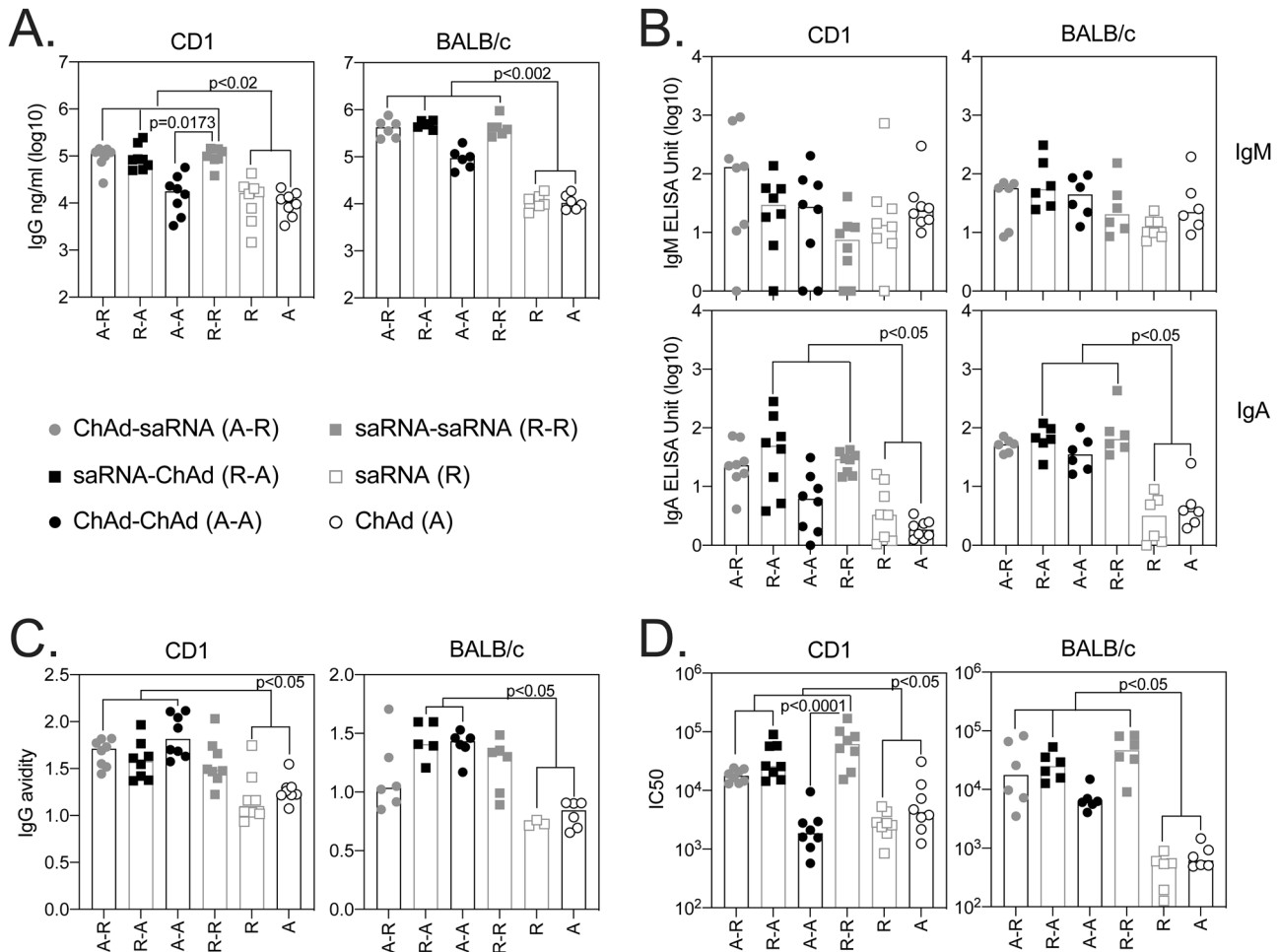

**Fig. 1 Antibody responses following ChAd and saRNA vaccination.** Antibody responses were measured in the serum of CD1 ($n = 8$) and BALB/c ($n = 6$) mice collected 3 weeks after the final immunization. Graphs show SARS-CoV-2 spike-specific IgG (**A**), IgM and IgA (**B**), and IgG avidity (**C**) measured by ELISA, and SARS-CoV-2-pseudotyped virus neutralization (IC50) (**D**). Individual mice are represented by a single data point; bars represent the median response in each group (CD1 $n = 8$; BALB/c $n = 6$). Data in each graph were analysed with a one-way ANOVA Kruskal–Wallis test followed by post hoc Dunn's multiple comparisons test to compare differences between vaccination groups; $p$-values indicate significant differences ($p < 0.05$) between groups.

with subsequent emergency licensure granted in some countries. Although each individual vaccine candidate has been thoroughly tested for safety and efficacy, there have been no studies reported to date, which have examined the safety, efficacy or any added benefit of mixed modality vaccinations. Given the real-world vaccination initiatives that are being progressed, there are scenarios wherein an individual receives a vaccine prime and a boost dose from different manufacturers or of different vaccine types. This current preclinical study examined the cellular and humoral immune responses in mice following vaccination with either the ChAdOx adenoviral vector or the saRNA in homologous or heterologous prime-boost combinations and strongly supports the need for clinical trial assessment of heterologous prime-boost regimens.

All prime-boost regimens elicited high levels of SARS-CoV-2 spike-specific antibodies with neutralization capacity and high avidity, levels that were greater than single vaccines alone. Heterologous vaccination regimens induced some of the highest antibody responses post vaccination, with neutralizing titres after heterologous prime boost at least comparable to or higher than those achieved after homologous vaccination with ChAdOx1 nCoV-19, whereas homologous saRNA and ChAd induced higher antibody responses than single-dose regimens, which is consistent with previous data in mice, pigs and non-human primates

(NHPs)[11,12]. We have demonstrated in NHPs that homologous vaccination with ChAdOx1 nCoV-19 results in protection against disease, with more recent data, in hamsters, demonstrating that a single immunization with ChAdOx1 nCoV-19 protects against disease induced by the variants of concern B.1.351 and B.1.1.7[14]. Importantly, in human clinical trials, strong enhancement of the antibody response was observed following a booster dose of ChAdOx1[15], with this regimen shown to be efficacious against SARS-CoV-2 disease in late-stage clinical trials[8], whereas recent real-world data, in elderly frail people, has demonstrated vaccine effectiveness after the first dose of ChAdOx1 nCoV-19 at 80.4% (95% confidence interval 36.4–94.5) with broadly similar effectiveness measured after RNA (Pfizer) vaccination[16].

Although there are no defined correlates of protection from clinical trials, rhesus macaques studies have demonstrated a clear role for neutralizing antibodies and also CD8+ T cells in protecting against disease[17]. In agreement, human studies have demonstrated neutralizing antibodies and T cells play an important role in preventing severe disease and augmenting recovery from coronavirus disease 2019 (COVID-19)[18]. Both vaccine modalities also elicited high numbers of antigen-specific T cells, which were further increased in the heterologous regimens. The majority of the IFNγ ELISpot response was directed against the S1 spike protein, in particular the first half (317 AA), which does

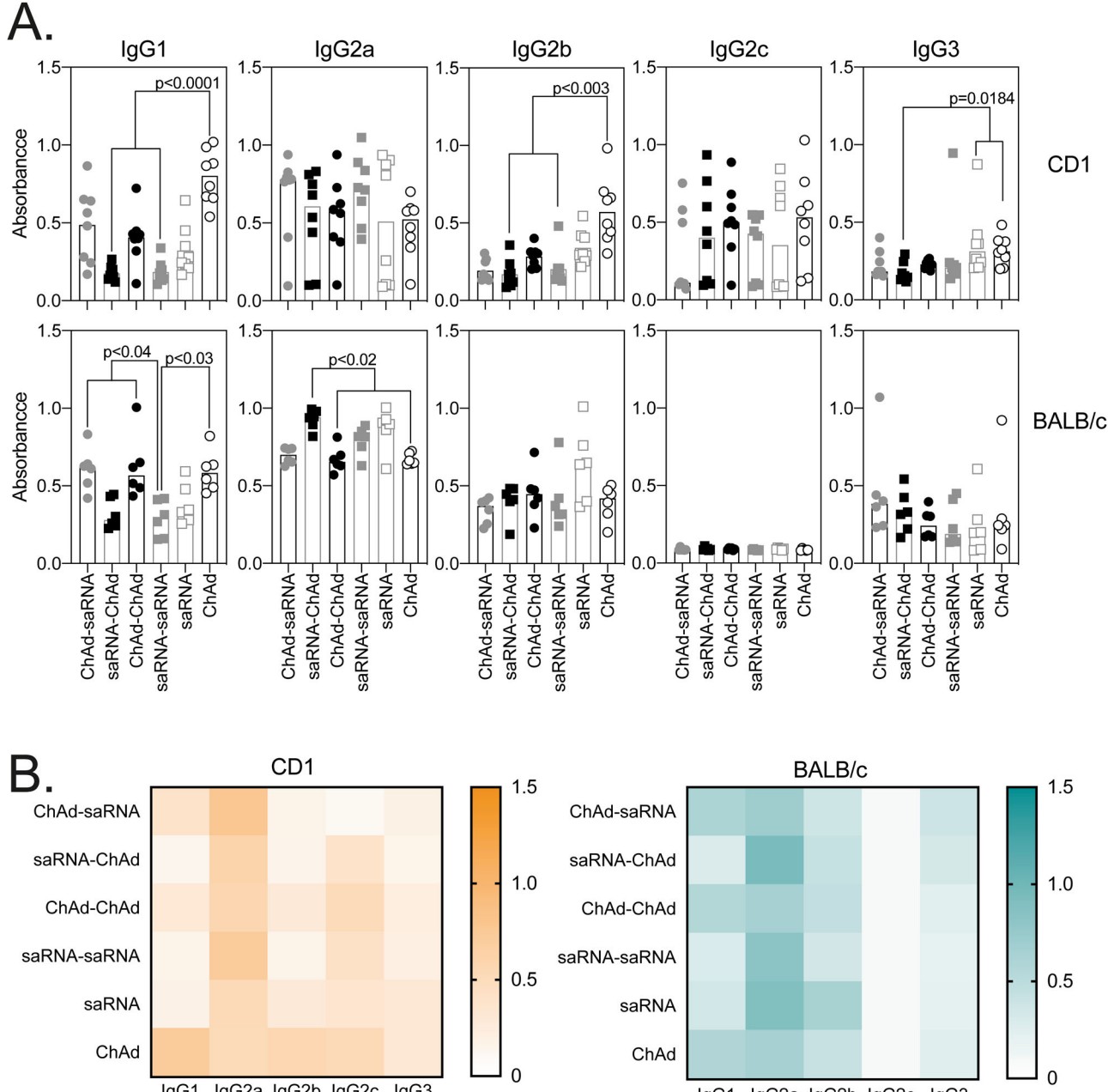

**Fig. 2 SARS-CoV-2 spike-specific IgG subclasses following ChAd and saRNA vaccination.** For detection of IgG subclasses in the serum of CD1 ($n = 8$) and BALB/c ($n = 6$) mice, each sample was diluted to 1 IgG ELISA Unit. Graphs show optical density measured against each IgG subclass where individual data points were expressed as an OD and shown here as scatter dot plots with bars showing the median (**A**), followed by the heatmap summary representation with median response in each group to each IgG subclass (**B**). Individual mice are represented by a single data point, bars represent the median response in each group (CD1 $n = 8$; BALB/c $n = 6$) with serum collected 3 weeks after the final vaccination. Data in each graph was analysed with a one-way ANOVA Kruskal–Wallis followed by a post hoc Dunn's multiple comparison test to compare differences between vaccination groups; $p$-values indicate significant difference ($p < 0.05$) between groups.

not include the receptor binding domain (RBD). The cell-mediated response was dominated by cytotoxic T cells, with heterologous regimens inducing higher frequencies and total numbers of antigen-specific CD8$^+$ T cells. Although CD4$^+$ T-cell responses were overall lower in frequency and number than CD8$^+$ T-cell responses, all vaccination regimens elicited CD4$^+$ T-cell responses of a Th1-type response, even in the Th2-biased BALB/c background, a response indicating that the potential for antibody-dependent enhancement and subsequent enhancement of respiratory disease caused by Th2-type lung immunopathology is reduced[19–21].

Vaccination regimens that induce a broad immune response (humoral and cell-mediated) will likely be the best option for long-term protection against COVID-19. It remains to be determined whether the higher antibody titres following heterologous vaccination regimens, as measured here, results in longer lived immunity with a broader humoral response. Critically, the logistical challenges of administering vaccines in a rapidly evolving landscape of mass vaccination schemes, coupled with limited global supply, underpins the need to generate data on mixing vaccine modalities. It will be important to clinically assess whether the mixed modality regimens have an altered or lessened

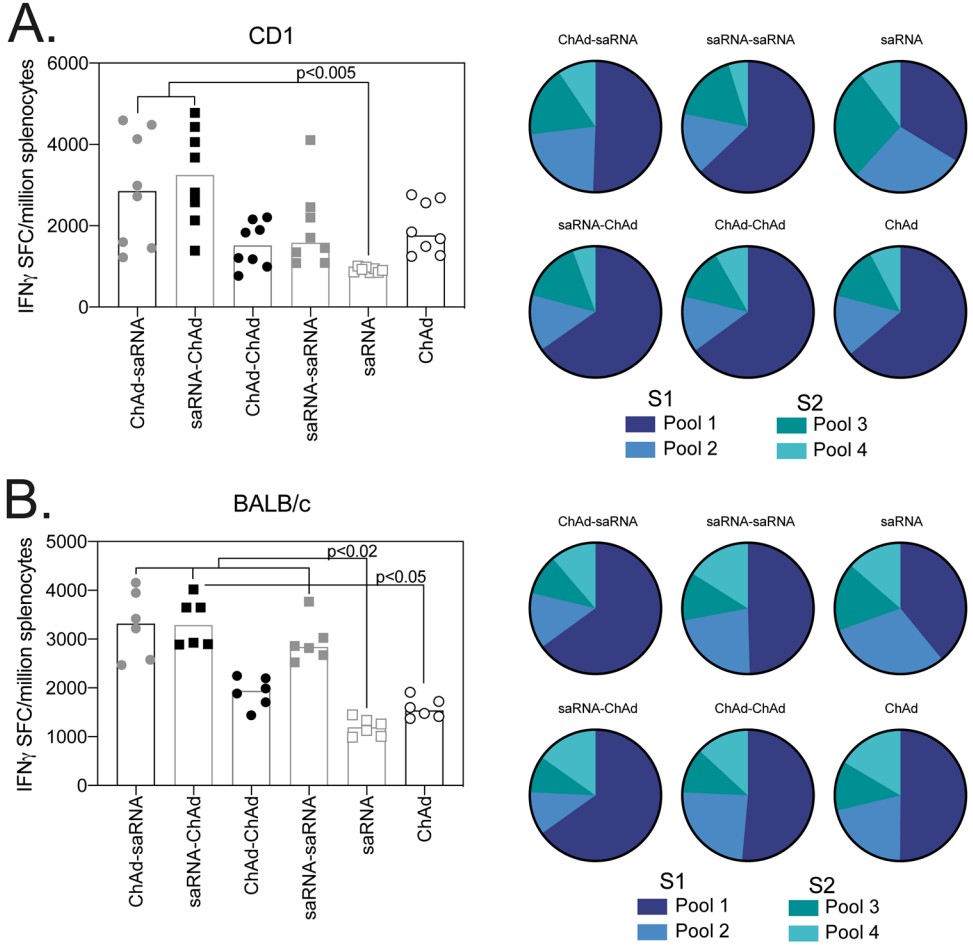

**Fig. 3 Breadth of T-cell response measured by ELISpot.** Graphs represent the total spike-specific IFNγ response (sum of peptide pools) measured in outbred CD1 ($n = 8$) (**A**) or inbred BALB/c ($n = 6$) (**B**) 3 weeks after the final vaccination. Pie charts represent the response to each peptide pool as a proportion of total response. Data points represent individual mice and bars represent the median response in each group. Data in each graph were analysed with a two-way ANOVA Friedman test and post hoc Dunn's multiple comparison to compare between vaccination regimens, $p$-values indicate significant difference ($p < 0.05$) between groups.

reactogenicity profile and, most importantly, the ability to augment protection against disease or onward transmission. Importantly, some of these questions will be addressed in a recent clinical study recruiting up to 820 participants to receive combinations of two different SARS-CoV-2 vaccines, which has been initiated in the United Kingdom (https://comcovstudy.org.uk/home). The data described herein reinforce the need for this and other clinical trials to assess the safety, immunogenicity and efficacy of heterologous vaccination regimens.

## Methods

**Ethics statement.** Mice were used in accordance with the UK Animals (Scientific Procedures) Act 1986 under project license number P9804B4F1 granted by the UK Home Office, with approval from the local Animal Welfare and Ethical Review Board at the University of Oxford. Age-matched animals were purchased from commercial suppliers as a batch for each experiment and randomly split into groups on arrival at our facility. Animals were group housed in individual ventilated cages (IVCs) under specific pathogen free (SPF) conditions, with constant temperature (20–24 °C) and humidity (45–65%) with lighting on a 13 : 11 light–dark cycle (7 a.m. to 8 p.m.). For induction of short-term anaesthesia, animals were anaesthetized using vaporized IsoFlo®. All animals were humanely killed at the end of each experiment by an approved Schedule 1 method.

**Animals and immunizations.** Outbred CD1Hsd:ICR (CD1) (Envigo) ($n = 8$ per group), Crl:CD1 (ICR) (Charles River) ($n = 8$ per group) and inbred BALB/cOlaHsd (BALB/c) (Envigo) ($n = 6$ per group) mice of 7 weeks of age were immunized intramuscularly (i.m.) in the musculus tibialis with $10^{8}$ infectious units

of ChAdOx1 nCoV-19[12], 1 µg saRNA[3] or received no prime vaccination. Mice were boosted (or primed for mice receiving only a single vaccination) i.m. with the relevant vaccine candidate 4 weeks later. All mice were killed 3 weeks after the final vaccination with serum and spleens collected for analysis of humoral and cell-mediated immunity.

**Pseudotype virus neutralization assay.** A human immunodeficiency virus (HIV)-pseudotyped luciferase-reporter-based system was used to assess the neutralization ability of sera from vaccinated mice. In brief, CoV S-pseudotyped viruses were produced by co-transfection of 293T/17 cells with a HIV-1 gag-pol plasmid (pCMV-Δ8.91, a kind gift from Professor Julian Ma, St George's University of London), a firefly luciferase-reporter plasmid (pCSFLW, a kind gift from Professor Julian Ma, St George's University of London) and a plasmid encoding the S protein of interest (pSARS-CoV2-S) at a ratio of 1 : 1.5 : 1. Virus-containing medium was clarified by centrifugation and filtered through a 0.45 µm membrane 72 h after transfection and subsequently aliquoted and stored at −80 °C. For the neutralization assay, heat-inactivated sera were first serially diluted and incubated with virus for 1 h and then the serum–virus mixture was transferred into wells preseeded with $Caco_2$ cells. After 48 h, cells were lysed and luciferase activity was measured using Bright-Glo Luciferase Assay System (Promega). The half maximal inhibitory concentration (IC50) neutralization was then calculated using GraphPad Prism (version 8.4). Statistical analyses were performed on log-transformed data.

**Antigen-specific IgG ELISA.** The antigen-specific IgG titres in mouse sera were assessed by a semi-quantitative ELISA. MaxiSorp high binding ELISA plates (Nunc) were coated with 100 µL per well of 1 µg/mL recombinant SARS-CoV-2 protein with the pre-fusion stabilized conformation in phosphate-buffered saline (PBS). For the standard IgG, three columns on each plate were coated with 1 in 1000 dilution each of goat anti-mouse κ- and λ-light chains (Southern Biotech).

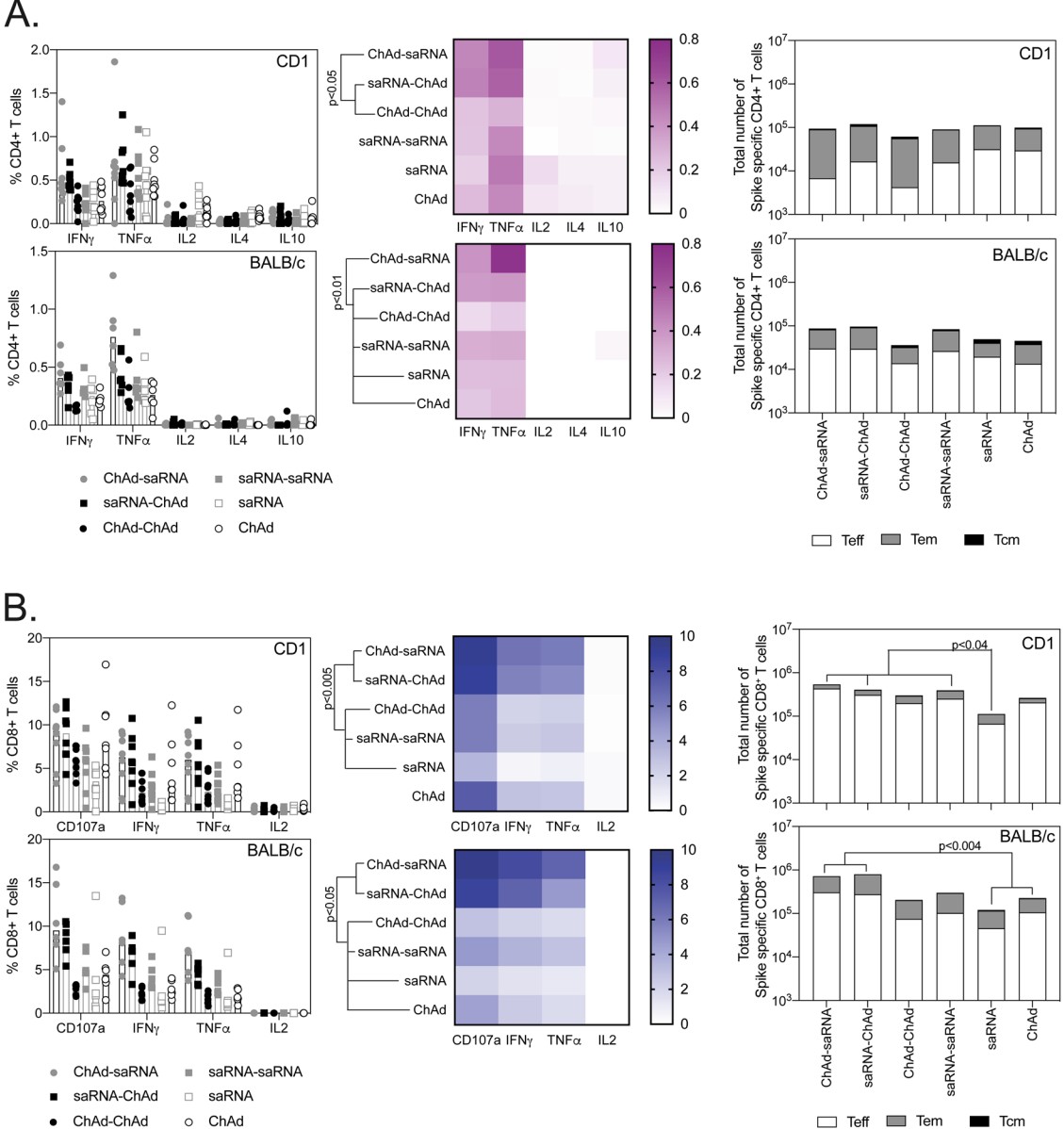

**Fig. 4 Phenotype of the T-cell response following vaccination.** CD1 ($n = 8$) and BALB/c ($n = 6$) splenocytes collected 3 weeks after the final vaccination were stimulated for 6 h with pools of overlapping SARS-CoV-2 peptides prior to staining for effector and memory T cells markers and intracellular cytokines. **A** Graphs show the frequency of the spike-specific CD4+ T-cell responses (left) in CD1 (top panel) and BALB/c mice (bottom panel); heatmaps (middle) show the proportion of the response producing each cytokine and total number of antigen-specific cells of a T effector (Teff), T effector memory (Tem) or T central memory (Tcm) phenotype (right). **B** Graphs show the frequency of the spike-specific CD8+ T-cell responses (left) in CD1 (top panel) and BALB/c mice (bottom panel); heatmaps (middle) show the proportion of the response producing each cytokine and total number of antigen-specific cells of a T effector (Teff), T effector memory (Tem) or T central memory (Tcm) phenotype (right). Data points indicate individual mice; bars represent the median response in each group. Total numbers of each population are displayed in Supplementary Fig S3C. Data in each graph were analysed with a two-way ANOVA comparing the effect of vaccination regimen and cytokine production or T-cell phenotype, followed by a post hoc Tukey's multiple comparison test to compare between vaccination regimens; p-values showing overall differences between vaccination groups ($p < 0.05$) are indicated on the graph.

After overnight incubation at 4 °C, the plates were washed four times with PBS–Tween 20 0.05% (v/v) and blocked for 1 h at 37 °C with 200 μL per well blocking buffer (1% bovine serum albumin (w/v) in PBS–Tween-20 0.05%(v/v)). The plates were then washed and the diluted samples or a fivefold dilution series of the standard IgG added using 50 μL per well volume. Plates were incubated for 1 h at 37 °C, then washed and secondary antibody added at 1 in 2000 dilution in blocking buffer (100 μL per well), and incubated for 1 h at 37 °C. After incubation and washes, plates were developed using 50 μL per well SureBlue TMB (3,3′, 5,5′-tetramethylbenzidine) substrate and the reaction stopped after 5 min with 50 μL per well stop solution (Insight Biotechnologies). The absorbance was read on a Versamax Spectrophotometer at 450 nm (BioTek Industries). Statistical analyses were performed on log-transformed data.

**Antigen-specific Isotype ELISA**. MaxiSorp plates (Nunc) were coated with 50 μl of 2 μg/mL or 50 μl of 5 μg/mL SARS-CoV-2 full-length spike (FL-S) protein overnight at 4 °C for detection of IgG (250 ng/well) or IgM and IgA (500 ng/well), respectively, prior to washing in PBS/Tween (0.05% v/v) and blocking with Blocker Casein in PBS (Thermo Fisher Scientific) for 1 h at room temperature (RT). Standard positive serum (pool of mouse serum with high-endpoint titre against FL-S protein), individual mouse serum samples, negative and an internal control (diluted in casein) were incubated for 2 h at RT for detection of specific IgG or 1 h at 37 °C for the detection of specific IgM or IgA. Following washing, bound antibodies were detected by addition of a 1 in 5000 dilution of alkaline phosphatase (AP)-conjugated goat anti-mouse IgG (Sigma-Aldrich), 1 in 5000 dilution of anti-mouse IgM (Abcam) or 1 in 1000 dilution of anti-mouse IgA (Southern Biotech)

for 1 h at RT, and addition of *p*-Nitrophenyl Phosphate, Disodium Salt substrate (Sigma-Aldrich). An arbitrary number of ELISA units (EUs) were assigned to the reference pool and optical density values of each dilution were fitted to a four-parameter logistic curve using SOFTmax PRO software. EUs were calculated for each sample using the optical density values of the sample and the parameters of the standard curve. IgM limit of detection was defined as 2 EUs and IgA limit of detection set as 6 EUs. All data were log-transformed for statistical analyses.

**Antigen-specific IgG subclass ELISAs.** MaxiSorp plates (Nunc) were coated with 50 μL of 2 μg/mL per well of SARS-CoV-2 FL-S protein overnight at 4 °C prior to washing in PBS/Tween (0.05% v/v) and blocking with Blocker Casein in PBS (Thermo Fisher Scientific) for 1 h at RT. For detection of IgG subclasses, all serum samples were diluted to 1 total IgG EU and incubated at 37 °C for 1 h prior to detection with AP-conjugated anti-mouse IgG subclass-specific secondary antibodies IgG1 (1 in 4000, Southern Biotech), IgG2a (1 in 4000, Southern Biotech), IgG2b (1 in 4000, Southern Biotech), IgG2c (1 in 4000, Southern Biotech) or IgG3 (1 in 1000, Abcam) incubated for 1 h at 37 °C. The results of the IgG subclass ELISA are presented using optical density values.

**Avidity ELISA.** Anti-SARS-CoV-2 spike-specific total IgG antibody avidity was assessed by sodium thiocyanate (NaSCN)-displacement ELISA. Nunc MaxiSorp ELISA plates (Thermo Fisher Scientific) coated overnight at 4 °C with 2 μg/well SARS-CoV-2 FL-S protein diluted in PBS were washed with PBS/Tween (0.05% v/v) and blocked for 1 h with 100 μl per well of Blocker Casein in PBS (Thermo Fisher Scientific) at 20 °C. Test samples and a positive control serum pool were diluted in blocking buffer to 1 total IgG EU and incubated for 2 h at 20 °C. After washing, increasing concentrations of NaSCN (Sigma-Aldrich) diluted in PBS were added and incubated for 15 min at 20 °C. Following another wash, bound antibodies were detected by addition of 1 in 5000 dilution of AP-conjugated goat anti-mouse IgG (Sigma-Aldrich) for 1 h at RT and addition of *p*-Nitrophenyl Phosphate, Disodium Salt substrate (Sigma-Aldrich). For each sample, concentration of NaSCN required to reduce the OD405 to 50% of that without NaSCN (IC50) was interpolated from this function and reported as a measure of avidity.

**Antigen-specific B-cell staining.** Spike, RBD and a decoy NANP$_9$C (repeat region from *Plasmodium falciparum* CSP protein) tetramers were prepared in-house by mixing biotinylated proteins with streptavidin-conjugated flurochromes (A488, A647 or r-PE) in a 4:1 molar ratio and incubating on ice for 30 min. Splenocytes were stained with Spike-PE and RBD-A647 at a final concentration of 0.04 mM, whereas a decoy tetramer NANP$_9$C-A488 was used at a final concentration of 0.4 mM. Splenocytes were stained with Live-Dead Aqua and Fc block (anti-CD16/32 mAb, Clone 93, 1 in 50) prior to staining with the antibody cocktail containing NANP$_9$C-Alexa488, GL7-PerCPCy5.5 (Clone GL7, 1 in 100), CD138 BV421 (Clone 281-2, 1 in 100), CD95-BV605 (Clone SA367H8, 1 in 100), CD4-BV650 (Clone GK1.5, 1 in 200), CD279-BV711 (Clone 29 F.1A12, 1 in 100), CD19-BV780 (6D5, 1 in 200), RBD-A647, IgD-A700 (Clone 11-26 c.2a, 1 in 100), IgM-APCCy7 (Clone 11/41, 1 in 100), Spike-PE, CD38-PECY5 (Clone 90, 1 in 200), CD69-PeCy7 (Clone H1.2F3, 1 in 100), CD45R-BUV395 (Clone RA3-6B2, 1 in 200) and CD3-BUV496 (Clone 145-2C11, 1 in 200) antibodies purchased from BioLegend, BD or Invitrogen. Antigen-specific B cells were identified by gating on LIVE/DEAD negative, size (Forward Scatter (FSC)-A vs. side scatter (SSC)), doublet negative (FSC-H vs. FSC-A), CD45RA$^+$, CD19$^+$ and NANP-A488$^-$, RBD-A647$^+$ and Spike-PE$^+$ followed by staining for GC or class-switched B cells (Supplementary Fig. S2A). The total number of cells was calculated by multiplying the frequency of each population, expressed as a percentage of total lymphocytes, by the total number of lymphocytes counted for each individual mouse spleen sample.

**ELISpot and ICS staining.** Spleen single-cell suspension were prepared by passing cells through 70 μM cell strainers and Ammonium-Chloride-Potassium solution (ACK) lysis prior to resuspension in complete media. Splenocytes were stimulated 15mer peptides (overlapping by 11) spanning the length of SARS-CoV-2 spike protein and tpa promoter, with peptide pools subdivided into peptides spanning the S1 and S2 region of spike (Supplementary Table S1). For analysis of IFNγ production by ELISpot, splenocytes were stimulated with two pools of S1 peptides (pools 1 and 2) and two pools of S2 peptides (pools 3 and 4) (final concentration of 2 μg/mL) on hydrophobic PVDF-membrane ELISpot plates (Millipore) coated with 5 μg/mL anti-mouse IFNγ (AN18). After 18–20 h of stimulation at 37 °C, IFNγ spot forming cells were detected by staining membranes with anti-mouse IFNγ biotin (1 mg/mL) (R46A2) followed by streptavidin-AP (1 mg/mL) and development with AP conjugate substrate kit (BioRad, UK). Spots were enumerated using an AID ELISpot reader and software (AID).

For analysis of intracellular cytokine production, cells were stimulated at 37 °C for 6 h with 2 μg/mL pool of S1 (ELISpot pools 1 and 2) or S2 (ELISpot pools 3 and 4) peptides (Supplementary Table S1), media or positive control cell stimulation cocktail (containing PMA-Ionomycin, BioLegend), together with 1 μg/mL Golgi-plug (BD) and 2 μl/mL CD107a-Alexa647 (Clone 1D4B). Following surface staining with CD3-A700 (Clone 17A2, 1 in 100), CD4-BUV496 (Clone GK1.5, 1 in 200), CD8-BUV395 (Clone 53-6.7, 1 in 200), CD44-BV780 (Clone IM7, 1 in 100), CD62L-BV711 (Clone MEL-14, 1 in 100), CD69-PECy7 (Clone H1.2F3, 1 in 100) and CD127-BV650 (Clone A7R34, 1 in 100) or CD127-APCCy7 (Clone A7R34 1 in 100), cells were fixed with 4% paraformaldehyde and stained intracellularly with TNFa-A488 (Clone MP6-XT22, 1 in 100), IL2-PerCPCy5.5 (Clone JES6-5H4, 1 in 100), IL4-BV605 (Clone 11B11, 1 in 100), IL10-PE (Clone JES5-16E3, 1 in 100) and IFNγ-e450 (Clone XMG1.2, 1 in 100) diluted in Perm-Wash buffer (BD). Sample acquisition was performed on a Fortessa (BD) and data analysed in FlowJo V10 (TreeStar). An acquisition threshold was set at a minimum of 5000 events in the live CD3$^+$ gate. Antigen-specific T cells were identified by gating on LIVE/DEAD negative, doublet negative (FSC-H vs. FSC-A), size (FSC-A vs. SSC), CD3$^+$, CD4$^+$ or CD8$^+$ cells, and each individual cytokine or 'cytokine positive' comprising a combination of 'CD107a or IFNγ, or TNFα or IL2, or IL4 or IL10' populations. Cytokine-positive responses are presented after subtraction of the background response detected in the corresponding media-stimulated control sample for each mouse and summing together the response detected to each pool of peptides. Teff cells were defined as CD62L$^{low}$ CD127$^{low}$, Tem cells defined as CD62L$^{low}$ CD127$^{hi}$ and T central memory cells defined as CD62L$^{hi}$ CD127$^{hi}$ (Supplementary Fig. S3B). The total number of cells was calculated by multiplying the frequency of the background-corrected population (expressed as a percentage of total lymphocytes) by the total number of lymphocytes counted in each individual spleen sample.

**Statistical analysis.** All graphs and statistical analysis were performed using Prism v9 (Graphpad). For analysis of vaccination regimen against a single variable (e.g., IgG level), data were analysed with a one-way analysis of variance (ANOVA; Kruskal–Wallis) followed by post hoc Dunn's multiple comparison test. For analysis of vaccination regimen against multiple variables (e.g., each individual cytokine or T-cell subset), the data were analysed with a two-way ANOVA, where a significant difference was observed; a post hoc analysis was performed to compare the overall effect of vaccination regimen. In graphs where a significant difference was observed between multiple vaccine groups, the highest *p*-value is displayed on the graph. All data displayed on a logarithmic scale was log$_{10}$ transformed prior to statistical analysis (EUs, neutralization titres, total cell numbers).

**Reporting summary.** Further information on research design is available in the Nature Research Reporting Summary linked to this article.

**Data availability**
The data that support the findings of this study are available within the article and its Supplementary Information files, or are available from the corresponding author upon reasonable request. Source data are provided with this paper.

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

## Acknowledgements

We thank D. Pulido, H. Davies and F. Donnellan for provision of spike and RBD proteins, BMS staff for animal husbandry and A. Worth, J. Furze, M. Mykhaylyk and R. Evans for facilities support. This report is an independent research funded by the National Institute for Health Research (UKRI Grant Ref: MC_PC_19055, NIHR Ref: COV19 OxfordVacc-01) and through support made by philanthropic donors to the University of Oxford's COVID-19 Research Response Fund. The views expressed in this publication are those of the author(s) and not necessarily those of the funders.

## Author contributions

A.J.S., M.U., S.B.-J., C.B., K.S., K.M., A.B., D.W. and P.M.K. performed experiments. A.J.S., M.U., H.S., C.G., E.A. and A.T. performed animal procedures and/or sample processing. A.J.S., S.B.J., K.H. and P.M.K. analysed data. A.J.S., T.L., P.M.K., R.S. and S.G. designed the study. A.J.S., S.B.J., P.M.K. and T.L. wrote the manuscript. All authors reviewed the final version of the manuscript.

## Competing interests

S.C.G. is the co-founder and board member of Vaccitech (collaborators in the early development of this vaccine candidate), and is named as an inventor on a patent covering the use of ChAdOx1-vectored vaccines and a patent application covering this SARS-CoV-2 vaccine. T.L. is named as an inventor on a patent application covering this SARS-CoV-2 vaccine and was consultant to Vaccitech. P.M.K. and R.J.S. are co-founders and R.J.S. is a board member of VaxEquity and VacEquity, and are named inventors on a patent application covering the SARS-CoV-2 saRNA vaccine candidate.
