## [Peer Review File · Nature Communications]

Reviewers' Comments:

Reviewer #1:

Remarks to the Author:

In this manuscript, Spencer et al. explore the immunogenicity of SARS-CoV-2 vaccine heterologous prime/boost strategies in two different mouse strains using ChAd and saRNA encoding the Spike antigen. The heterologous prime/boost concept is interesting, and worth being pursued, albeit not novel as it has already been described using ChAd and MVA for example. This is the first report exploring the benefit of using ChAd and saRNA, however, the data provided here fail to demonstrate any meaningful superiority of heterologous prime/boost compared to the homologous regimen. As acknowledged by the authors, the data presented in this manuscript confirm that prime/boost and single dose regimens induced Th1 Spike-specific humoral and cellular immune responses as previously published. In its current version, this manuscript does not provide enough information to clearly understand how the data was obtained.

Major concerns

- The title must be toned down as only spike specific CD8 T cells might be increased after heterologous prime/boost
- It is not clear if mice receiving a single ChAd or saRNA vaccine got saline as a boost 4 weeks later and when were these animals sacrificed ... 3 weeks after the first injection or at 7 weeks like the groups receiving prime/boosts?
- Identification of SARS-CoV-2 spike-specific B cells by flow cytometry: the description provided by the authors in the methods section and in FigS2 are not detailed enough to understand how the data presented in FigS2B were obtained and if the statements in lines 96-99 are supported
- FigS2: there are several concerns with these data and figure
 - o please provide the entire gating strategy for B-cell characterization
 - o FigS2 A: not clear what are top and bottom dot-plots
 - o FigS2 B: how was the total number calculated?
- Identification of SARS-CoV-2 spike-specific T cells by flow cytometry: not enough detailed information is provided to understand how the % of CD4 or CD8 producing any of the cytokines shown in Fig4 A and B are calculated. The middle heatmap is unclear as of what it is representing and to which cytokine(s) do the p-values refer to.
- FigS3 B: there are several concerns with the gating strategy presented.
 - o Time vs FSC-A is the least sensitive/discriminative combination to identify perturbations of the signal during acquisition. Ideally, Time should be checked against each fluorescence channel. At a minimum Time vs SSC-A should be used.
 - o FSC-H vs FSC-A is the least discriminative combination to identify doublets. The Height parameter does not change between a singlet or a doublet, while Area and Width are directly proportional. SSC-A vs SSC-W is the most sensitive combination to resolve singlets from doublets (See Doublet Discrimination. UWCCC Flow Cytometry Laboratory, <https://cancer.wisc.edu/research/resources/flow/>)
 - o What is CD44 used for in this gating strategy? If used after the CD4 and CD8 gates, one could gate out the naïve T cells.
 - o It is not clear how Cytokine+ T cells are gated/obtained. Based on quadrants? Please specify.
 - o I understand from the workflow that memory T-cell populations are further assessed from the cytokine+ cells with CD62L vs CD127 plots. If this is the case, please explain the presence of naïve CD62L+CD127- cells. Otherwise, please provide a more comprehensive explanation how spike-specific memory T cells are gated/calculated
 - o A bi-exponential scale on the x-axis would help better resolve the CD127- populations
- FigS3C: how was the total number calculated?
- Line 279, same concerns as for T cells regarding gating out doublets

Minor comments

- Line 102, T-cell mediated immunity ... would be more precise and appropriate here as the authors

just reported about B-cells responses in the previous paragraph.

- Lines 274 – 277, when are CD138, CD279, CD38, CD69, CD3 used and what for?
- Line 280 is size discrimination really using FSC-H vs SSC? See comment below for line 304 for T cells
- Line 296, CD69 is cited here but not mentioned anywhere else. What was it used for?
- Line 304, the description of size (FSC-H vs SSC) does not correspond to the gating strategy shown in FigS3B (FSC-A vs SSC-A)
- Line 308 should read "T effector memory cells defines as CD62L-CD127+)..."

Reviewer #2:

Remarks to the Author:

The paper by Spencer et. al. describes the effect of heterologous vaccination with saRNA and adenoviral vectored vaccines against SARS-CoV-2 in mice. Although this is a well written manuscript and the detailed description of the effect on the immune response is important, the fact that addition of a booster compared to a single vaccination is beneficial is overall not novel. Since the AZD1222 was developed and approved for emergency use as prime-boost vaccine, it is not clear what is the relevance of comparing a single dose regimen to a heterologous one and why is the focus of this paper is not whether a heterologous vaccination can be advantageous over the approved homologous regimen. The novelty and impact of such data comparing the heterologous to the approved homologous regimen may improve significantly the strength of this paper and must be addressed thoroughly.

In addition, the superiority of the immune response in the heterologous vaccination regimens as compared to the homologous or single ones, as stated by the authors, can be much more convincing if a SARS-CoV-2 challenge animal model will be used following vaccination (i.e. hACE2 K-18 mice).

Minor points to address

Introduction section:

- An explanation on why the saRNA was chosen (and for example not the mRNA emergency approved platforms (i.e. Pfizer, Moderna)) should be added (pros and cons).

Results section:

- Fig S1A. Please check whether the ssRNAX2 is not statistically significant compared to the single doses.

- Please add in line 97 the mouse strain (BALB/c).

- The B cells data is not discussed. What is the importance of this data?

- Please specified what are the differences between the S1 (pool 1-pool 2) and S2 (pool 3-pool 4).

- Please specified for what Teff, Tem and Tc stands for.

- The last 2 sentences (starting from line 119) are confusing. What is the difference between them.

Please clarify.

Methods section:

- Line 227, please specified what is FL-S.

- Lines 284-5, what are the S1 and S2 pools made of?

- Please add a detailed statistical analysis section.

Figure legends:

- Please add to each figure legend the number of animals and the relevant time points.

- Figure 4 – The heatmaps statistics is not clear. Does the comparison is for a specific cytokine or is it for all off them? For the total number of cells, is it for the entire population or for a specific cell type (i.e. Teff, Tem or Tcm)?

Reviewer #1 (Remarks to the Author):

In this manuscript, Spencer et al. explore the immunogenicity of SARS-CoV-2 vaccine heterologous prime/boost strategies in two different mouse strains using ChAd and saRNA encoding the Spike antigen. The heterologous prime/boost concept is interesting, and worth being pursued, albeit not novel as it has already been described using ChAd and MVA for example. This is the first report exploring the benefit of using ChAd and saRNA, however, the data provided here fail to demonstrate any meaningful superiority of heterologous prime/boost compared to the homologous regimen. As acknowledged by the authors, the data presented in this manuscript confirm that prime/boost and single dose regimens induced Th1 Spike-specific humoral and cellular immune responses as previously published. In its current version, this manuscript does not provide enough information to clearly understand how the data was obtained.

We thank the reviewer for their comments and agree heterologous vaccine route is worthy of clinical investigation. The majority of the currently licensed vaccines use two dose regimens. However, exploratory clinical analysis has demonstrated vaccine efficacy from day 22 and up to 12 weeks after a single vaccination with ChAdOx1 nCoV-19 to be 76% (59.3-85.9) (PMID: 33617777). While recent real world data, in elderly frail people, has demonstrated vaccine effectiveness after the first dose of ChAdOx1nCoV-19 at 80.4% (95% CI 36.4-94.5) with broadly similar effectiveness measured after RNA (Pfizer) vaccination. There is therefore a broad interest in comparing heterologous vaccination regimens to both single shot and homologous regimens. We have modified the text and title to reflect this.

Major concerns

•*The title must be toned down as only spike specific CD8 T cells might be increased after heterologous prime/boost*

Our original title was based on the data showing increased heterologous responses relative to single doses, we have now changed the title of the paper as requested so the focus of the paper is more presentation of immunogenicity of the heterologous regimen.

•*It is not clear if mice receiving a single ChAd or saRNA vaccine got saline as a boost 4 weeks later and when were these animals sacrificed ... 3 weeks after the first injection or at 7 weeks like the groups receiving prime/boosts?*

We thank the reviewer for pointing this out, vaccinations were synchronised so that all mice were sacrificed on the same day to ensure immunological analysis was performed together. Therefore the mice that received only one injection were left untreated at the first vaccination timepoint and received the single dose of vaccine at the same time as the pre-primed mice received their booster dose, all mice were then sacrificed a further 3 weeks later. We have added text to the methods to make this clearer to the reader.

•*Identification of SARS-CoV-2 spike-specific B cells by flow cytometry: the description provided by the authors in the methods section and in FigS2 are not detailed enough to understand how the data presented in FigS2B were obtained and if the statements in lines 96-99 are supported*

We apologise for the lack of sufficient detail, we have amended as needed.

•*FigS2: there are several concerns with these data and figure*
o please provide the entire gating strategy for B-cell characterization
A full gating strategy has now been provided.

o FigS2 A: not clear what are top and bottom dot-plots

We thank the reviewer for noting this, the top panel is a representative plot from one of the vaccinated animal, lower panel is a naïve mouse. This is to demonstrate that there is a detectable population of antigen specific cells in vaccinated mice when compared to naïve.

o FigS2 B: how was the total number calculated?

For calculation of the total number of cells, each subset is expressed as a percentage of total lymphocytes (final size gate) and then multiplied by the total number of lymphocytes counted in the spleen sample of each animal. We also check relative proportion of dead cells, and lymphocytes in each individual sample to ensure consistency across samples. In our experience, whether the final population is a frequency of lymphocytes or total cells acquired on the flow cytometer does not make a significant impact on the total number of cells, or impact on difference between groups of vaccinate mice.

•*Identification of SARS-CoV-2 spike-specific T cells by flow cytometry: not enough detailed information is provided to understand how the % of CD4 or CD8 producing any of the cytokines shown in Fig4 A and B are calculated. The middle heatmap is unclear as of what it is representing and to which cytokine(s) do the p-values refer to.*

We have expanded the text and added more detail as to how the population of cytokine positive cells is identified. We have clarified in the text what is represented in the heatmap and the p-value.

•*FigS3 B: there are several concerns with the gating strategy presented.*

o *Time vs FSC-A is the least sensitive/discriminative combination to identify perturbations of the signal during acquisition. Ideally, Time should be checked against each fluorescence channel. At a minimum Time vs SSC-A should be used.*

We thank the reviewer for their comments, we manually check each sample. With the FSC and SSC voltages that we run on our instrument, we have found that Time vs FSC give the greatest discrimination of aberrant acquisition events over the collection period and enables exclusion of the first and last seconds of sample acquisition.

o *FSC-H vs FSC-A is the least discriminative combination to identify doublets. The Height parameter does not change between a singlet or a doublet, while Area and Width are directly proportional. SSC-A vs SSC-W is the most sensitive combination to resolve singlets from doublets (See Doublet Discrimination. UWCCC Flow Cytometry Laboratory, <https://cancer.wisc.edu/research/resources/flow/>)*

We thank the reviewer for their insightful comments, while we agree that SSC may now be a more routine method of gating out doublets, the high frequency of cytokine positive cells observed in mice (relative to human, NHP and ferret samples), a tight doublet gate makes minimal impact on the final percentage or total number of cytokine positive cells, particularly as we further sub-gate on CD4s and CD8s.

o *What is CD44 used for in this gating strategy? If used after the CD4 and CD8 gates, one could gate out the naïve T cells.*

While we have not gated for CD44+ or CD44hi cells, by displaying CD44 against CD3 it enables clearer identification activated CD3 cells that have decreased CD3 surface expression. These cells would be hard to discriminate from CD3 negative cells in a CD3 vs FSC plot for instance.

o *It is not clear how Cytokine+ T cells are gated/obtained. Based on quadrants? Please specify.*

We thank the reviewer for their comments and have included a description in the methods. Cytokine positive cells are gated Boolean gate made as a combination of “IFN γ or TNF α or IL2 or IL4 or IFN γ or IL10 or CD107a”. This population is expressed as a percentage of CD4 or CD8 cells and antigen S1 or S2 specific frequency calculated by subtracting frequency of this cytokine population detected in the “media” stimulated sample.

o *I understand from the workflow that memory T-cell populations are further assessed from the cytokine+ cells with CD62L vs CD127 plots. If this is the case, please explain the presence of naïve CD62L+CD127- cells. Otherwise, please provide a more comprehensive explanation how spike-specific memory T cells are gated/calculated*

As the plot is showing the Boolean gate containing “IFN γ or TNF α or IL2 or IL4 or IL10 or CD107” there is a large amount of non-specific cells that will make it into the gating strategy prior to subtraction of the frequency in media stimulated cells. The small population of

CD62L+ CD127- is simply noise in this plot. We overcome this by taking each T cell subset as a percentage of CD4 or CD8 cells, and then subtract the background frequency of this population detected in the media stimulated samples. If we take only the IFN γ + population of cells and carry on with memory populations we detect very similar number of each T cell population (as the cytokine positive response is dominated by IFN γ + cells), but not wanting to bias the analysis of the memory response to only Th1 cells, we chose to use a broader cytokine positive gate for the subsequent analysis of memory T cell populations.

o A bi-exponential scale on the x-axis would help better resolve the CD127- populations

We thank the reviewer for their comment and agree that while bi-exponential display can sometimes help to resolve expression, because CD127 is expressed on all CD4 and CD8 cells and expression decreases post-activation, we find bi-exponential display compresses the positive and negative populations together and makes gating difficult. The optimal way to then gate CD127 is to compare expression on antigen specific cells relative to the total CD4 or CD8 population without bi-exponential display.

•FigS3C: how was the total number calculated?

The population is expressed as a percentage of total lymphocytes and multiplied by the total number of lymphocytes counted in the organ to calculate the total number of cells.

•Line 279, same concerns as for T cells regarding gating out doublets

We thank the reviewer for their comments and take on board their comments, as mentioned above with the user preferred voltages and cytometer settings of our machine and limited number of doublet that are created with our flow staining protocol, we find the FSC-A vs FSC-H is sufficient to gate out doublets. In addition, we also use a non-specific B cell stain to further reduce the potential of non-specific cells being gated into our antigen specific population.

Minor comments

•Line 102, T-cell mediated immunity ... would be more precise and appropriate here as the authors just reported about B-cells responses in the previous paragraph.

We agree and have now edited the text accordingly.

•Lines 274 – 277, when are CD138, CD279, CD38, CD69, CD3 used and what for?

CD69, CD279 and CD3 were included in this panel as exploratory markers for T follicular helper cells gating, while CD138 and CD38 were included to potentially measure plasma B cells. We chose to only present data on antigen specific B cells, as the other markers did not add to the dataset, but choose to include all antibodies used in the panel for full disclosure.

•Line 280 is size discrimination really using FSC-H vs SSC? See comment below for line 304 for T cells

We thank the reviewer for noting this typo, text should state FSC-A not FSC-H

•Line 296, CD69 is cited here but not mentioned anywhere else. What was it used for?

CD69 was included in the staining panel to investigate cells of a tissue resident memory phenotype in the spleen following Adenovirus vaccination. As it was not the focus of the study and percentage is small, we have not included this analysis in the paper, but mention the inclusion of the CD69 antibody for full disclosure.

•Line 304, the description of size (FSC-H vs SSC) does not correspond to the gating strategy shown in FigS3B (FSC-A vs SSC-A)

We thank the reviewer again for noting this typo, it should state FSC-A not FSC-H.

•Line 308 should read “T effector memory cells defines as CD62L-CD127+)...”

We thank the reviewer for noting this and have amended the text.

Reviewer #2 (Remarks to the Author):

The paper by Spencer et. al. describes the effect of heterologous vaccination with saRNA and adenoviral vectored vaccines against SARS-CoV-2 in mice. Although this is a well written manuscript and the detailed description of the effect on the immune response is important, the fact that addition of a booster compared to a single vaccination is beneficial is overall not novel. Since the AZD1222 was developed and approved for emergency use as prime-boost vaccine, it is not clear what is the relevance of comparing a single dose regimen to a heterologous one and why is the focus of this paper is not whether a heterologous vaccination can be advantageous over the approved homologous regimen. The novelty and impact of such data comparing the heterologous to the approved homologous regimen may improve significantly the strength of this paper and must be addressed thoroughly. In addition, the superiority of the immune response in the heterologous vaccination regimens as compared to the homologous or single ones, as stated by the authors, can be much more convincing if a SARS-CoV-2 challenge animal model will be used following vaccination (i.e. hACE2 K-18 mice).

We thank the reviewer for their positive review of our manuscript. We have modified the text to facilitate comparison of heterologous vaccination regimens to homologous regimens. However given the real-world data demonstrating effectiveness of a single dose of ChAdOx1nCoV-19 or RNA-based approaches, there is a broad interest in comparing heterologous vaccination regimens to both single shot and homologous regimens.

Minor points to address

Introduction section:

- An explanation on why the saRNA was chosen (and for example not the mRNA emergency approved platforms (i.e. Pfizer, Moderna)) should be added (pros and cons).

We thank the reviewer for their comments, large Pharma are frequently reluctant to collaborate on such studies, particularly when it involves direct comparison of vaccine modalities and typically these studies can only be performed after licensure. At present, even though emergency use access has been granted in a number of countries for Pfizer or Moderna vaccines, there is such short supply it would be unethical and impractical to attempt to use those vaccines for the work described here. It was through our strong collaborations with other academic researchers at Imperial college and a mutual willingness to compare vaccine platforms that we were able to investigate heterologous studies with saRNA vaccines. The saRNA vaccine has been funded by the UK government and CEPI and has been tested in early stage human trials studies in the UK.

Results section:

- Fig S1A. Please check whether the ssRNAX2 is not statistically significant compared to the single doses.

We thank the reviewer for their comments, RNA x 2 is only statistically significant compared to single doses in CD1 mice.

- Please add in line 97 the mouse strain (BALB/c).

Thank you for noting this, text added.

- The B cells data is not discussed. What is the importance of this data?

We have now added an additional sentence to discuss.

- Please specified what are the differences between the S1 (pool 1-pool 2) and S2 (pool 3-pool 4).

We have now added text to clarify and also added a table listing all the peptides in pools 1 to 4 into the supplementary material.

- Please specified for what Teff, Tem and Tc stands for.

We apologise for the omission of this detail, abbreviations are now explained in the main text in addition to the methods.

- The last 2 sentences (starting from line 119) are confusing. What is the difference between them. Please clarify.

We thank the reviewer for pointing this out and have changed the text accordingly.

Methods section:

- Line 227, please specified what is FL-S.

We thank the reviewer for noting this, FL-S is the abbreviation for full-length spike and we have clarified this in the text.

- Lines 284-5, what are the S1 and S2 pools made of?

These peptides pools comprise of all the overlapping peptides spanning the length of S1 or S2, a table has been added to the manuscript for more detail.

- Please add a detailed statistical analysis section.

A statistical analysis section has now been included in the methods.

Figure legends:

- Please add to each figure legend the number of animals and the relevant time points.

We thank the reviewer for pointing this out and have now ensured these details are in all figure legends.

- Figure 4 – The heatmaps statistics is not clear. Does the comparison is for a specific cytokine or is it for all off them? For the total number of cells, is it for the entire population or for a specific cell type (i.e. Teff, Tem or Tcm)?

We apologise for the ambiguity, the statistical analysis presented in the heat-map analyses compares overall difference between vaccination regimens, taking the difference in frequency of each single cytokine between groups into consideration. In this analysis, vaccination regimen and each cytokine are analysed as a variable. For simplicity, the post-hoc test compares vaccination regimen as a whole, instead of difference between each cytokine for and each vaccination regimen.

A similar analysis was performed for T cells subsets, comparing overall effect of vaccination regimen when taking into account each population as a variable, with the post-hoc test comparing vaccination regimens, not vaccination regimens for each individual population.

Reviewers' Comments:

Reviewer #1:

Remarks to the Author:

In this manuscript, Spencer et al. explore the immunogenicity of SARS-CoV-2 vaccine heterologous prime/boost strategies in two different mouse strains using ChAd and saRNA encoding the Spike antigen. The manuscript has been revised to better focus on the immunogenicity of the heterologous regimen.

Most of the comments have been adequately addressed in the revised version.

Sylvie Bertholet

REVIEWERS' COMMENTS

Reviewer #1 (Remarks to the Author):

In this manuscript, Spencer et al. explore the immunogenicity of SARS-CoV-2 vaccine heterologous prime/boost strategies in two different mouse strains using ChAd and saRNA encoding the Spike antigen. The manuscript has been revised to better focus on the immunogenicity of the heterologous regimen.

Most of the comments have been adequately addressed in the revised version.

Sylvie Bertholet

We thank the reviewer for their re-review of our manuscript.